

# Brief communication: Vehicle routing problem and UAV application in the post-earthquake scenario

Marco Cannioto[1], Antonino D'Alessandro[2], Giosuè Lo Bosco[1], Salvatore Scudero[2], Giovanni Vitale[2]

[1]Dipartimento di Matematica e Informatica, Università degli studi di Palermo, Palermo, Via Archirafi 34, 90123, Italy
[2]Istituto Nazionale di Geofisica e Vulcanologia, Centro Nazionale Terremoti, Rome, Italy

*Correspondence to*: Salvatore Scudero (salvatore.scudero@ingv.it)

**Abstract.**. In this paper we simulate a Unmanned Aerial Vehicle's (UAV) recognition after a possible case of diffuse damage after a seismic event in the town of Acireale (Sicily, Italy).Given a set of sites(84 relevant buildings) and the range of the UAV, we are able to find the number of vehicles to employ and the shortest survey path. The problem of finding the
shortest survey path is an operational research problem called Vehicle Routing Problem (VRP) whose solution is known to be computationally time-consuming. We used the Simulated Annealing (SA) heuristic that is able to provide stable solutions in relatively short computing time. We also examined the distribution of the cost of the solutions varying the depot on a regular grid in order to assess the best area where to execute the survey.

## 1 Introduction

In the very last decade some strong earthquakes struck central Italy (L'Aquila, 2009; Emilia, 2012; and Amatrice, 2016) causing relevant social and economical consequences. In the immediate post-event it is crucial to perform the fastest recognition of the damaged area in order to rescue as much people is possible and to assess and map the damage scenario.

At the same time, in the last decade, the Unmanned Aerial Vehicle (UAV) greatly diffused because of their versatility(Gomez and Purdie, 2016):UAVs are able to self navigate to complete a programmed mission, or to be remotely
controlled, are prompt to perform tasks, having no obstacles like any other ground vehicle. For all these features UAVs are becoming the safest, fastest, and most reliable mean to perform observation, surveying and mapping in various types of emergency implicating large inaccessible areas (Griffin, 2004; Chou et al., 2010;Obanawa et al., 2014; Giordan et al., 2015, 2017;Dominici et al., 2016; Jurecka and Niedzielski, 2016;Silvagni et al., 2016).Readiness and reliability are key aspects in the post-earthquake scenario, when the first problem to deal with is to detect the *hot spots*: the areas where threatened people
needing assistance or rescue are supposed to be and the areas where the key infrastructures (e.g. hospitals, schools, etc.) are.UAV recognitions are also essential to find the accessible and safe routes for the rescue vehicles.

The tasks that the UAV (or a group of UAVs) has to accomplish should be optimized to avoid waste of time and resources. Moreover, in such emergency scenario, some questions arise: how many UAVs employ? What will be the autonomy? How long does the survey will take? Where is the best taking-off place? What is the best route for every UAV in order to



minimize the travel distance and the surveying time? To solve all these questions we will apply to the UAV the concepts of the *Vehicle Routing Problem* (VRP) in a possible real scenario (Agatz et al., 2015; Golden et al., 2008; Wang et al., 2017; Yu et al., 2017). In the case study we do not have to consider the capacity of transport since we will not carry any goods; the UAVs will just have a survey task and have to visit each site (given its geographical coordinates) as shortest as possible,

taking into account their autonomy and return back to the depot: such approach of the VRP is called "Distance-Constrained" (DVRP) in which the main task is to minimize the total length of the routes.

Our case study is the town of Acireale (Sicily, Italy) where more than 52,000 people live. The city is located along the eastern coast of Sicily, just on the flank of Mt. Etna volcano. In historic time the town of Acireale experienced harsh damages, victims, and even almost complete destruction, after several strong earthquakes (Azzaro et al., 2000; Rovida et al.,

2016) and according to the Italian seismic hazard map it is among the areas where the highest peak ground accelerations (up to 0.25 g) are expected (Ordinanza PCM, 2006). For these reasons, a pilot urban dense seismic network realized with Micro Electro-Mechanical Systems (MEMS) sensors is being installed in the town for early warning purposes and rapid damage assessment after a seismic event (D'Alessandro, 2014; D'Alessandro et al., 2014; D'Alessandro, 2016). MEMS accelerometers are deployed in the most sensitive buildings such as hospitals, schools, and all the facilities devoted to the

public security (D'Alessandro and D'Anna, 2003). In this paper we will test a routing algorithms in order to find the best solutions for a complete recognition of the 84 selected sites within the town of Acireale.

## 2 Methods and materials

The specific problem we want to solve belongs to the category of operational research problems known as Vehicle Routing Problems (VRP). In detail VRP is a combinatorial optimization and integer programming problem that generalizes the

Travelling Salesman Problem (TSP). From the computational point of view, the solution of this problem is identified as NP-hard (Garey and Johnson, 1990) i.e. informally that it is not known an exhaustive solution to the problem that can be performed in polynomial time. Due to such computational limitations, we have decided to provide a solution by an incomplete method for optimization called Simulated Annealing.

### 25  2.1 The Simulated Annealing for optimization problem solution

The Simulated Annealing (SA) method is based on the analogy with the so-called *annealing process* i.e. the heating up and subsequent cooling down process of a solid that, consequently, freezes into a minimum energy structure. Kirkpatrick et al. (1983) introduced the method as an optimization technique for combinatorial problems, but it takes inspiration from later work done by Metropolis et al. (1953) and Pincus (1970). The annealing process starts providing high temperatures to the

solid. The effect is that the atoms of the solid assume high energy states so that they are more able to arrange themselves. This atomic energy reduces while the temperature is reduced until a minimum of energy state is reached where the solid





reaches the crystal structure. During the whole cooling process, it is really important to apply a slowly cooling: a very quickly one does not involve a minimum energy state of the solid, and some kind of irregularity and defects appear suddenly in the crystal structure.

In the case of thermal equilibrium at a temperature $T$, the Boltzmann distribution gives the probability $P(T,s)$ that the system

is in a given configuration:

$$P(T,s) = \frac{e^{-E(s)/kT}}{\sum_{q \in S} e^{-E(q)/kT}} \tag{1}$$

Where $E(s)$ is the energy of the configuration and $k$ is the Boltzmann's constant and $S$ indicates the set of all possible configurations. A system of particles in thermal equilibrium at a given temperature $T$ can be simulated using a technique

developed by Metropolis et al, (1953). Suppose at time $t$, the system is in configuration $q$, and a new candidate configuration $r$ is generated randomly at time $t+1$. The configuration $r$ is accepted or rejected according to the ratio $p$ between the probabilities of being in $r$ and in $q$:

$$p = \frac{P(T,r)}{P(T,q)} = e^{-(E(r)-E(q))/kT} \tag{2}$$

In particular, if $p>1$ ($E(r)<E(q)$), then $r$ is accepted as the new configuration at time $t+1$. If $p <= 1$ ($E(r)>E(q)$), $r$ is accepted

as the new configuration with probability $p$. This process involves that configurations with higher energy can be achieved. Moreover, it has been shown that as time $t \to \infty$, the probability that the system is in a given configuration $s$ is $P(T,s)$, regardless of starting configuration.

Dealing with a generic global optimization problem, the analogy is done by considering the states of the solid $S$ as the feasible space and the energy of the states $E(s)$ as the objective function to optimize. The solution of the problem is the

minimum energy state. The SA algorithm is iterative and it uses the relaxation technique by Metropolis described above as strategy for finding the solution. The implementation of the SA algorithm for the solution of a specific optimization problem, involves the following fundamental choices:

i) *Representation of the solution space*: By a mapping $S \to R$ of the feasible space $S$ of the optimization problem into a new space $R$, a new representation of the solution is provided.

ii) *Cost function*: The cost function $C$ measures the quality of a given solution.

iii) *Transition mechanism*: To move from one state to the next, a transition mechanism $P$ that slightly modifies the current solution is needed.

iv) *Cooling schedule*: The temperature at initial state, the temperature updating rule and the number of iterations to be carried out at each step of the cooling are fundamental for the cooling process of simulated annealing. A

stopping criterion for the termination of the search process is needed.

SA's major advantage over other incomplete methods is the ability to avoid becoming trapped in local minima. The algorithm adopts a particular random search which accepts changes that can locally improve or worsen the function to optimize. It can be shown that, for any given finite problem, the probability that the simulated annealing algorithm





terminates with the global optimal solution approaches 1 as the annealing schedule is extended. This theoretical result is, however, not particularly helpful, since the annealing time required to ensure a significant probability of success will usually exceed the time required for a complete search of the solution space.

In order to implement the *SA* solution for the *VRP*, we represent the solution by a permutations of $I + J - 2$ where $I$ indicates the number of sites and $J$ the number of UAVs. This representation is such that all the $J - 1$ elements $K$ of $s$ verifying $s(K) > I - 1$ delimitates $L(j), j = 1,..,J$ subset of elements in $s$ which represent the routes of each UAV. Taking into consideration that each UAV must take off and land from the same depot $d$, the final route to consider will be $\{d, L(j), d\}$. The starting solution is set as a random permutation, and at each iteration of the algorithm the maximum distance $MD$ run by the UAVs, and their total distance $TD$ are computed. Note that it can occur that the algorithm

sets a solution that assigns a route to an UAV whose total distance is greater than its autonomy. To avoid this case, the binary vector of the feasible UAVs is computed so that $F(j) = 0$ means that the drone $j$ could not complete its route. Finally the cost of a solution is so defined :

$$C = \left(\frac{TD+9*MD}{10}\right) * \left(1 + 5 * \mu(F)\right) \qquad (3)$$

Where $\mu(F)$ is the mean of the values in $F$. The goal of the SA algorithm is to minimize the function $C$. The adopted transition mechanisms consist in adopting one of this operation, chosen at random:

    *i)*        *Swap*: swap two elements in *s* at random positions *i* and *j*.

    *ii)*      *Reversion*: reverse the entire content of the vector *s* bounded by two random indices *i* and *j*.

*iii)*     *Move*: move a random element $s(i)$ into a random position $j$

The transition mechanism is applied m times, in order to find a set of $N(s)$ of $m$ neighbors of a solution $s$. Among them, the best one $q$ in term of minimum cost in $N(s)$ is selected as new solution. The initial temperature has been set to 100, at each iteration the temperature is updated following the rule $T_{new} = 0.98 * T$. The maximum number of iteration has been set to 1000. All these parameters, together with the integer values in the cost function C, have been chosen after a trial and error

phase.

## 2.2 Study area and vehicle

The procedure aims to find the best survey solutions for a set of sites (given their geographical coordinates), identifying the best depot for the UAVs and minimizing the number of vehicles. The surveying sites are the vertices of the accelerometric network installed in the town of Acireale (Sicily, Italy) and some other noteworthy buildings (D'Alessandro, 2016), for a

whole of 84 sites (Fig. 1).





The selected UAV is commercial and low-cost called Phoenix (http://www.twodots.it/prodotti/droni/falcon/). It is able to communicate with an operative center located up to 20 km away, accomplish pre-programmed missions bypassing eventual obstacles, and executes real-time variations of route ordered from the operative center. Phoenix is equipped with a gimbal camera, visual positioning system, micro-USB port, micro-SD slot, obstacle detection system, LEDs, four engines, a

telemetry system and other systems able to provide information about the base-vehicle connection and the battery charge status. The UAV is able to reach speed of 20 m/s and the battery ensures an energetic autonomy of about 30 minutes, therefore in ideal conditions it could travel as far as 36 km.

## 3 Results

For the selected UAV the maximum considered travel distance is 30 km, but taking into account any possible unfavorable

atmospheric phenomena, the relevant difference in elevation between some sites, and a margin of safety, we repeated the tests with progressive increasing step of 3 km (3, 6, 9, 12, 15, 18, 21, 24, 27, and 30 km).

None solution was found for autonomy of 3 km. For autonomy of 6 km the solution is found with the use of 3 UAVs whereas for all the other autonomy values the solution consider the enrollment of 2 UAV. All the solutions have comparable travel distance but the estimated best depot varies. Among the performed solutions we show the ones relative to the shortest

and longest autonomy: 6 km and 30 km respectively (Fig. 2).The tested method showed a good velocity of convergence reaching a stable solution within about 30 s for all the selected autonomies. Table 1 summarizes, for each UAV range autonomy, the number of UAV, the cost and computing time of the solution, the distance travelled by the fleet, and the average distance travelled by every single UAV. The values of the cost, computing time, and are distances are almost constant anddo not show any relevant dependence with the different range autonomy, except for autonomy of 6 km. In this

case, because a third UAV is considered, the values of all the parameters are sensibly out of the range of the other considered autonomies.

Immediately after an earthquake some areas will not be accessible from the ground, therefore we should consider a possible set of depots for the UAVs. For this reason, after the first phase in which we calculated the best depot, in a following phase we executed the algorithm from all the vertexes of a regular grid.

In this second phase we thus evaluated the behaviour of the algorithm, given the autonomy, varying the depot within a 10 x 10 grid ($\sim 8$ km$^2$) of depot superimposed onto the survey area (Fig. 1). The grid size has been chosen to have a density comparable to the survey sites. We obtained the distribution of the cost for the solution executed at every grid point for the different considered autonomy. Furthermore, the results are interpolated and the maps for the shortest and the longest autonomy (i.e. 6 km and 30 km) are showed in Fig. 3 where the lighter shades indicate area where the cost is lower (i.e.

solutions are better) than areas with darker shades. The blue circles indicate the depot where the algorithm did not provide any solution, while among all the feasible solutions represented by the red circles, the green circle indicates the best solution (lowest cost). Solutions for all the 100 vertexes of the depots grid were found with autonomy greater than 9 km and the cost





pattern is somehow comparable since the solution with the lowest costs are located in the inner part of the grid. However, for some values of autonomy this pattern is not evident and a patchy map of the cost is obtained.

## 4 Discussion and conclusions

The SA algorithm was able to find solutions with only 2 UAVs (i.e. best solutions) even though solutions up to 10 UAVs were explored. Considering a more populous UAV fleet, the computing time to get a stable solution steadily increases. Conversely, increasing the number of UAV, the cost of the solution sharply decreases at the beginning, and later is almost constant (Table 2).

The algorithm is able to provide the best solution in term of number of UAV and selection of the depot given the autonomy range of the vehicle but we also took into account the eventual impossibility to operate the survey starting form the best depot in the post-earthquake scenario. We thus examined the spatial distribution of the cost for the solution executed from a grid of depots (Fig. 3). Moreover, we computed an interpolation of the averaged cost values of all the solutions at the different autonomy values and for each grid vertex: the pattern of the distribution is concentric and the best solution are around the central part of the study area (Fig. 4).In a post-earthquake scenario, the depots from the inner part of the town will be considered first, if exploitable, and later the depots from the outskirt. Even though with greater costs, solutions from the outer depots are valid and the UAVs taking off from these depots will be able to accomplish their task.

The SA showed to be a good method for the routing problem in term of computing time and reliability of the solutions. Our results also indicate that the selected UAV type is suitable to perform the survey of the area and, finally, results suggest that, in equal conditions (i.e. same UAV type an number) it would be possible to perform survey over a wider area. Therefore the same procedure could be operational also during the planned expansion for the accelerometric network in the town of Acireale. At the same way it could be implemented in other areas in analogous post-earthquake, or, in general, post-emergency scenarios.

## Competing interests

The authors declare that they have no conflict of interest.

## Acknowledgements

The work presented in this paper was partly supported by the project MEMS funded by SIR-MIUR 2014 (grant RBSI14HL4E).





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



**Table1.UAV number, cost, computing time, and distances for the best SA solutions.**

| Autonomy | UAV number | Cost | Computing time (s) | Total distance (km) | Average distance (km) |
|---|---|---|---|---|---|
| 3 | - | - | - | - | - |
| 6 | 3 | 6 .2952 | 31 .0703 | 15.5446 | 5.1815 |
| 9 | 2 | 7 .9211 | 29 .1417 | 14.3876 | 7.1938 |
| 12 | 2 | 7 .8579 | 29 .0943 | 14.1802 | 7.0901 |
| 15 | 2 | 7 .9304 | 29 .1680 | 14.3926 | 7.1963 |
| 18 | 2 | 8 .0011 | 29 .1712 | 14.4301 | 7.216 |
| 21 | 2 | 8 .0591 | 29 .1685 | 14.6524 | 7.3162 |
| 24 | 2 | 7 .7269 | 29 .1657 | 14.048 | 7.024 |
| 27 | 2 | 8 .0356 | 29 .1776 | 14.6039 | 7.302 |
| 30 | 2 | 7 .9703 | 29 .2260 | 14.4767 | 7.2384 |





**Table 2. Cost solution varying autonomy and UAVs number.**

| Autonomy | UAV number | | | | | | | | |
|---|---|---|---|---|---|---|---|---|---|
| | 2 | 3 | 4 | 5 | 6 | 7 | 8 | 9 | 10 |
| 3 | - | - | - | - | - | - | - | - | - |
| 6 | - | 6 .295 | 5 .365 | 4 .942 | 4 .928 | 4 .937 | 4 .913 | 4 .798 | 4 .908 |
| 9 | 7 .921 | 6 .361 | 5 .359 | 5 .075 | 4 .848 | 4 .853 | 4 .892 | 4 .888 | 4 .944 |
| 12 | 7 .858 | 6 .3 | 5 .228 | 5 .092 | 5 .004 | 4 .803 | 4 .947 | 4 .839 | 4 .996 |
| 15 | 7 .93 | 6 .277 | 5 .492 | 4 .905 | 4 .915 | 4 .924 | 4 .884 | 4 .924 | 4 .846 |
| 18 | 8 .001 | 6 .483 | 5 .41 | 5 .104 | 4 .92 | 4 .846 | 4 .881 | 4 .905 | 5 .017 |
| 21 | 8 .059 | 6 .412 | 5 .326 | 5 .062 | 4 .963 | 4 .988 | 4 .982 | 4 .887 | 4 .864 |
| 24 | 7 .727 | 6 .089 | 5 .383 | 5 .037 | 4 .915 | 4 .834 | 4 .934 | 4 .836 | 4 .876 |
| 27 | 8 .036 | 6 .267 | 5 .299 | 5 .059 | 4 .831 | 4 .898 | 4 .886 | 4 .897 | 5 |
| 30 | 7 .97 | 6 .281 | 5 .445 | 4 .985 | 4 .977 | 4 .935 | 4 .952 | 4 .83 | 4 .891 |



**Fig. 1. Location map of the town of Acireale (left) and of the 84 survey sites (right).**

**Fig. 2. Examples for SA solutionsfor6 km autonomy (left), and 30 km autonomy (right).**

5 **Fig. 3. On the left column the map of depots and sites, and on the right column the isosurfaces showing the cost of the solutions for all the depots in the grid. The lighter the colour in an isosurface, the better the solutions that starts from the enclosed depots. Examples for 6 km autonomy (top), and 30 km autonomy (bottom).**

**Fig. 4. Interpolated average value of the cost of the solution for all the considered autonomies. The lighter the colour in an**
10 **isosurface, the better the solutions that starts from the enclosed depots.**





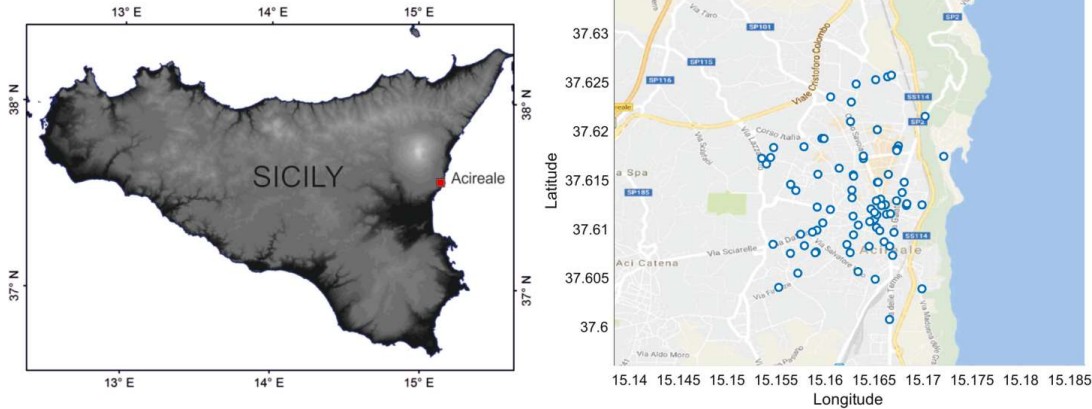

**Fig. 1**




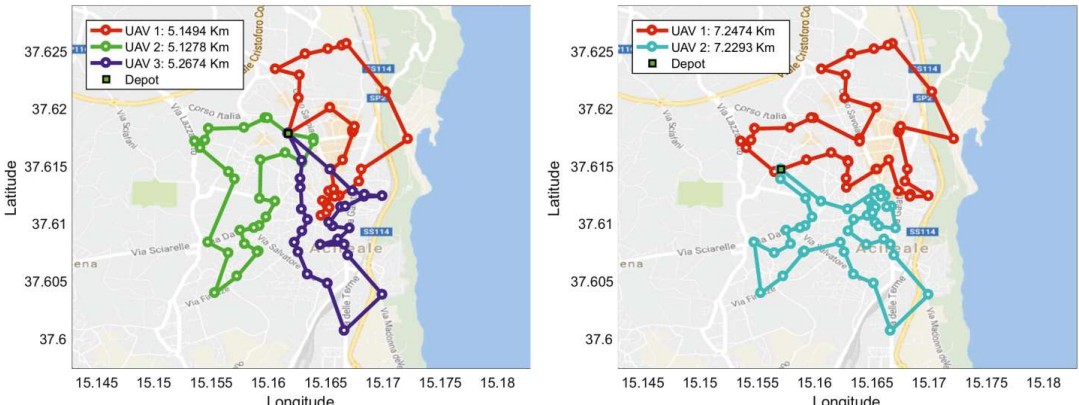

**Fig. 2**




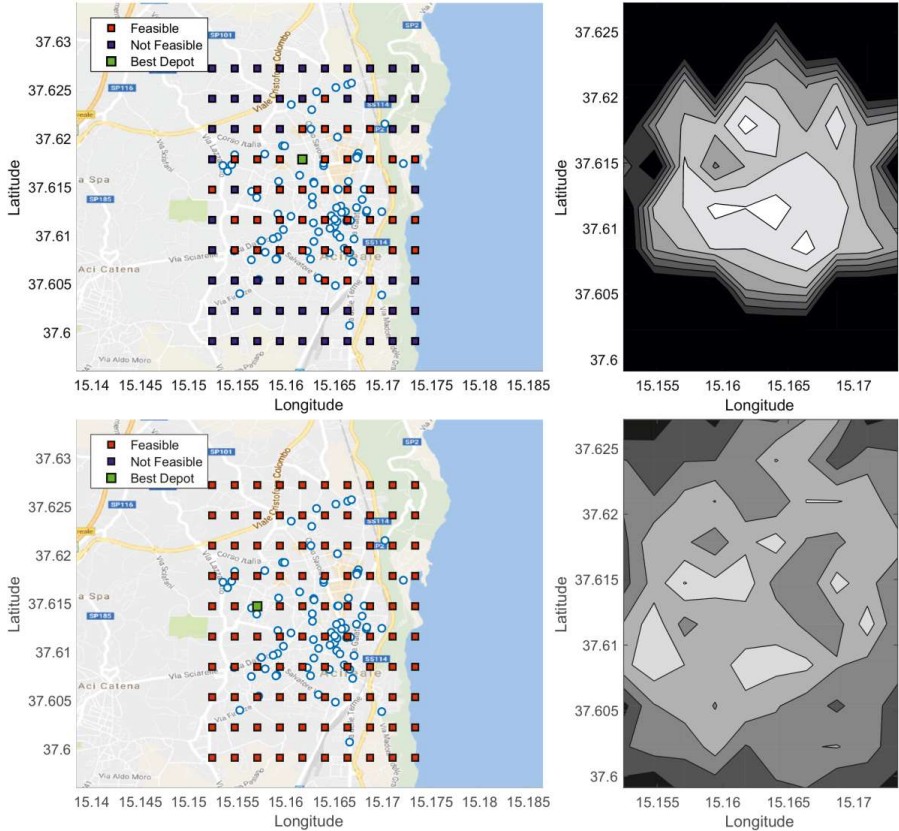

**Fig. 3**





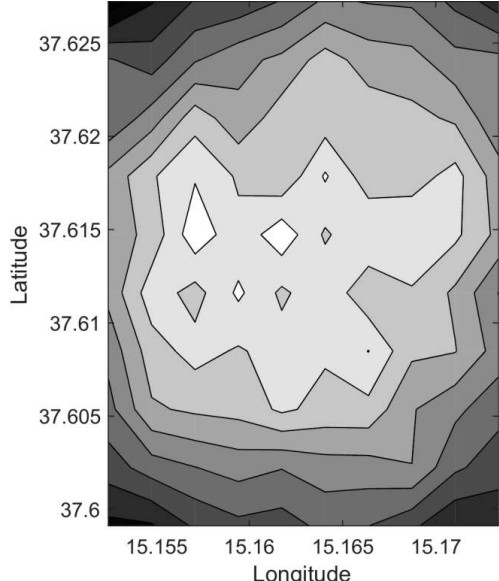

**Fig. 4**