# Peer review of "Brief communication: Vehicle routing problem and UAV application in the post-earthquake scenario"

_Natural Hazards and Earth System Sciences, 2017_

## Referee Comment (RC1) · Anonymous Referee #1 · 13 Jul 2017

General comments:

The brief communication with the title 'Vehicle routing problem and UAV application in the post-earthquake scenario' addresses the problem of finding the shortest survey path for a set of locations and UAVs in a post-disaster scenario. With the growing use of the UAV as main platform for such post-disaster surveys, the paper addresses an important problem related with the optimization of the use of such aircrafts. This optimization is both resource and time consuming, two critical factors in a post-disaster situation. The optimization is achieved using an optimization method denominated Simulated Annealing (SA) which aims for a global optimum instead of a precise local

optimum, reducing in this way, the computation time. The paper introduces the topic and the problem in a thorough way, considering it is a brief communication. The figures and tables shown illustrate well the results for the indicated objective. Conclusions are drawn based on the shown results.

Specific comments:

There is the need to point out the objective of the survey. While the cost of traveling from one point to the other is considered; there is no indication on the time taken to survey each building. The surveying time, per building, is important to consider, or at least address it in the paper. For example, given two points: a small school and a large hospital. To have an indication of the state of the building, these two places will need different times to be surveyed, hence influencing the global flight time. More critical if the objective is to have a thorough damage assessment of the building, hence needing oblique imagery and a greater amount of time for the survey.

Technical corrections:

The document, in general, is well written. Among others check line 18 and 19 for grammar/typos.

---

## Author Comment (AC1) · 18 Jul 2017

We thank the reviewer for taking time to read our paper and to give us his/her comments. The reviewer points out the omission of the UAVs' travel time in our paper. In our case-study the surveying time over each target site can considered equal, even though the different areal extension of the sites, because the aim of the survey is just to perform a fast visual inspection in the immediate post-earthquake in order to verify the state of damage of the buildings. At notable flying height (i.e. >50 m, the minimum ground distance for a safe survey in an urban area), the overflight above a site could be even neglected considering the optical characteristics of the UAV camera. We do

not intend to perform photogrammetric images for 3D reconstruction and damage assessment whose execution would be obviously dependent on the size of the building. In our case the damage assessment, at first step assessed by means of the accelerometric sensors installed at each site, is then verified by a rapid image comparison between the archived images and the observed scenario, rather than a time consuming (tens-of-hours to days) 3D reconstruction. However if different survey times should be considered, this could be easily addressed considering in the travel time between two sites, the surveying time over the second one. In the final version of the manuscript we will better indicate the objectives of the UAV mission (surveying as fast as possible) and make clearer our assumptions.

---

## Referee Comment (RC2) · Anonymous Referee #2 · 8 Sep 2017

General comments: The discussion paper addresses a relevant research topic which is of high importance to appropriately survey post-disaster environments. UAVs have proven to be a suitable tool to support search and rescue operations after natural hazards. However, the ad-hoc search of an efficient flight plan remains a challenge. The proposed method adequately addresses this task and reveals promising results. The study aims to find an optimal flight path that includes all relevant points of interest. For this, a method that follows the approach of "simulated annealing" is introduced. The method searches for the global optimization of a function and the authors adapted the algorithm to solve a vehicle routing problem in a real world scenario. The origins of the method as well as variances to the equation are well-described. The results show the

successful implementation of the method and are illustrated properly. In the second part of the results introduces a constraint in regard of the depots (take-off and landing point) which addresses an important element in a real post-disaster situation. Tables and figures support the written text and are well-presented.

Specific comments: Overall, the paper is well-structured and well-written. However, it is not clear which work has been done before to solve this kind of problems. How do other methods fail? Some sentences on the state of the art need to be inserted in order to define the objective of this discussion paper. Furthermore, the mission of the UAV is not is not really clear to me – does it follow the trajectory in a waypoint mode to capture nadir images or does it fly to each point of interest and make one picture? If the latter is the case – how can the damage be assessed if the UAV image shows only the roof? I think it would help to add some more explanations on the task of the UAV in subsection 2.2.

Technical corrections: The manuscript needs an English proofreading and corrections in grammar and style. Typos such as missing space character can be found throughout the paper and should be corrected.

---

## Author Comment (AC2) · 12 Sep 2017

We thank the reviewer for taking time to read our paper and to give us his/her comments. The main comment of the reviewer is about the choice of the methodology. Actually, the presented method (Simulated Annealing) has been chosen after being tested against another method, the Genetic Algorithm. The performances of the latter, in terms of computational time and robustness of the solutions, were clearly inferior with respect the Simulated Annealing. We think that is out of the objectives of the paper to present a comparisons between the two different explored methods, accordingly we focused just on the most promising algorithm. However, in the final version of the

manuscript we will make clear to the reader that the method has been selected after being tested against another one. Moreover, as well the previous reviewer, he points out the lack of the clear statement for the mission of the UAVs. In the final version of the manuscript we will better indicate the objectives of the mission (surveying as fast as possible) and make clearer our assumptions.

―――――――――――――――――

---

## Author Comment (AC3) · 18 Sep 2017

Dear Editor, please find attached the revised version of the manuscript titled "Brief communication: Vehicle routing problem and UAV application in the post-earthquake scenario". We modified the manuscript accordingly the reviewer's comments. We now better indicate the objectives of the UAV mission (surveying as fast as possible) and make clearer our assumptions (c.f. section 2.2). Moreover, we made clear to the reader that the method has been selected after being tested against another one. Minor flaws and typos have been fixed.

Please also note the supplement to this comment:
https://www.nat-hazards-earth-syst-sci-discuss.net/nhess-2017-192/nhess-2017-192-AC3-supplement.zip

[Figure]

**Fig. 1.**

[Figure]

**Fig. 2.**

[Figure]

**Fig. 3.**

[Figure]

**Fig. 4.**

---

## Author Response (AR1)

Dear Editor,

please find attached the revised version of the manuscript titled "Brief communication: Vehicle routing problem and UAV application in the post-earthquake scenario".

We modified the manuscript accordingly the reviewers' and editor's comments. We now better indicate the objectives of the UAV mission (surveying as fast as possible) and make clearer our assumptions (c.f. section 2.2).

Moreover, we made clear to the reader that the method has been selected after being tested against another one and we also explained in detail why the other method failed with respect the adopted one (c.f. section 2).

The reference list was updated.

Minor flaws and typos have been fixed.

[revised manuscript text omitted]

Fig. 1

[Figure]

**Fig. 2**

[Figure]

**Fig. 3**

[Figure]

**Fig. 4**